# Synthesis, Characterization, and Electrochemical Properties of La-Doped α-Fe_2_O_3_ Nanoparticles

**DOI:** 10.3390/nano12193346

**Published:** 2022-09-26

**Authors:** Hewei Wang, Peiyang Shi, Junxiang Ning

**Affiliations:** Key Laboratory for Ecological Metallurgy of Multimetallic Ores (Ministry of Education), School of Metallurgy, Northeastern University, Shenyang 110819, China

**Keywords:** pH value, La-doped α-Fe_2_O_3_ nanoparticle, electrochemical stability, corrosion resistance

## Abstract

La-doped α-Fe_2_O_3_ nanoparticles were synthesized by a hydrothermal method. The effects of pH value on the morphology, structure, and electrochemical stability of the La-doped α-Fe_2_O_3_ nanoparticles were investigated by X-ray diffraction, transmission electron microscopy, Fourier-transform infrared spectrum, and electrochemical methods. The results show that the La-doped α-Fe_2_O_3_ nanoparticles exhibit a uniform spherical morphology at pH = 6, and are agglomerated with a poor dispersion at pH = 4 and 8. The iron oxide lattice is distorted by the La-doping, which increases the Fe–O bond strength. The decreased Fe–O bond length and the increased Fe–O bond energy at pH = 6 improve the electrochemical stability of α-Fe_2_O_3_. The waterborne coating modified with La-doped α-Fe_2_O_3_ nanoparticles exhibits a steady corrosion resistance.

## 1. Introduction

Metal corrosion affects economic development and production safety [1]. Coating is an effective strategy to protect metals from corrosion, which has been widely used in the metal protection field [2,3]. Iron oxide is regarded as an environmentally friendly inorganic coating material due to its chemical stability, covering, and coloring properties [4,5,6,7,8,9]. However, restricted by its corrosion-prone behavior, the application of iron oxide as a coating is limited [10]. Nanotechnology can modulate the structure and properties of iron oxide, which plays an important role in improving the corrosion resistance [11,12]. However, relevant research is still limited.

Rare earth La has a special electronic layer, which can be used to inhibit corrosion. Qin et al. [13] found that La and Ce elements with a special 4f electronic structure could easily release the trapped electrons and form shallow traps, thereby prolonging the lifetime of photogenerated electron–hole pairs and improving the photocatalytic activity of TiO_2_. Ning et al. [14] synthesized Ce-doped α-Fe_2_O_3_ nanoparticles using a hydrothermal method. It was found that Ce-doping leads to the lattice distortion of α-Fe_2_O_3_, which not only enhances the Fe–O bond energy, but also increases the chemical stability of α-Fe_2_O_3_. Currently, research mainly focuses on the dielectric properties [15,16], magnetic properties [17,18,19,20,21], adsorption properties [22], and magnetothermal properties [23] of La-doped ferrite or other composites. However, La-doped iron oxide is not widely reported. Shan et al. [24] synthesized La-doped α-Fe_2_O_3_ nanotubes by electrospinning. The sensing performance of acetone is higher on La-doped α-Fe_2_O_3_ nanotubes than that on α-Fe_2_O_3_ nanotubes. Aghazadeh et al. [25] investigated the electrochemical performance of undoped and La-doped magnetite nanoparticles by cyclic voltammetry and galvanostatic charge–discharge test. The results showed that La doping improved the capacitance of iron oxide. Melo et al. [26] synthesized La-doped Fe_2_O_3_ pigment from a polymer precursor using the Pechini method. Ravinder et al. [27] adopted the sol–gel method to synthesize nano α-Fe_2_O_3_. It was found that the crystal size decreases with an increase in the La content, and the average magnetization intensity decreases when the Fe^3+^ is replaced by La^3+^. Raj [28] found that La^3+^ completely replaced Fe^3+^ by substitution in La-doped Fe_2_O_3_ particles, which influenced the shape, the size, the distribution, and the band gap of the crystals. Zan et al. [29,30,31] found that the prepared super-foldable C-web/FeOOH-nanocone and super-foldable composite electrode can enhance the functional requirements of flexible electronic materials and also expand the application prospects of iron-based materials, providing a new research direction for the application of La-doped α-Fe_2_O_3_. To our best knowledge, the corrosion resistance properties of La-doped iron oxide have not been reported.

In this work, La-doped α-Fe_2_O_3_ nanoparticles were synthesized by a hydrothermal method. The effects of pH value on the structure and morphology of α-Fe_2_O_3_ were studied. The structure of the La-doped α-Fe_2_O_3_ lattice was simulated by Materials Studio (MS) software, and the electrochemical stability of La-doped α-Fe_2_O_3_ nanoparticles was investigated. In addition, the La-doped α-Fe_2_O_3_ nanoparticles were applied to the waterborne coatings, and the corrosion resistance of the coatings was studied. This research was helpful to explore the structure and electrochemical properties of La-doped α-Fe_2_O_3_, and provided technical support for the development of corrosion-resistant waterborne coatings.

## 2. Experimental Section

### 2.1. Materials

La_2_(SO_4_)_3_ (AR, >99%), Fe_2_(SO_4_)_3_ (AR, >99%), NaCl (AR, >99%), NaOH (AR, >96%), and H_2_SO_4_ (AR, >98%) were purchased from Sino-Pharm (Chemical Reagent Co., Ltd., Shanghai, China). All chemicals were directly used without purification.

### 2.2. Preparation of La-Doped α-Fe_2_O_3_ Nanoparticles

The La-doped α-Fe_2_O_3_ nanoparticles were synthesized using a hydrothermal method. In a typical synthesis process, 1.7 g La_2_(SO_4_)_3_ and 120.0 g Fe_2_(SO_4_)_3_ were dissolved in 600 mL deionized water in a flask. The solution pH values were adjusted to 4, 6, and 8, respectively, with NaOH. The mixture was placed in a titanium alloy autoclave and treated at 160 °C for 1 h with a stirring speed of 200 rpm. Then, the autoclave was naturally cooled to room temperature. The products were washed with deionized water and ethanol several times. After drying at 120 °C for 6 h, La-doped α-Fe_2_O_3_ nanoparticles were obtained. The preparation of undoped α-Fe_2_O_3_ nanoparticles was similar to that of La-doped α-Fe_2_O_3_ nanoparticles, except that no La_2_(SO_4_)_3_ was added, which was presented as α-Fe_2_O_3_(u). The La-doped α-Fe_2_O_3_ nanoparticles synthesized at various pH values were presented as La-doped α-Fe_2_O_3_(x), where x denotes the pH value.

### 2.3. Preparation of La-Doped α-Fe_2_O_3_ Modified Waterborne Coatings

The synthesis parameters of the La-doped α-Fe_2_O_3_-modified waterborne coatings are shown in Table 1. Firstly, the H_2_O, dispersing agent, defoamer, coalescent, ethylene glycol, acrylic resin, and thickening agent were mixed, and stirred at 300 rpm for 10 min. Then, the solution pH value was adjusted to 8–9 with ammonia. Subsequently, the La-doped α-Fe_2_O_3_ nanoparticles were added to the solution. The pH value was maintained at 8–9 with ammonia. The mixture was stirred at 300 rpm for another 20 min. The viscosity range of the mixture was adjusted to 0.2–0.35 Pa·s by a viscometer (RTW−16, NEU Shenyang, China). The α-Fe_2_O_3_-modified waterborne coatings were presented as coating(u). The La-doped α-Fe_2_O_3_-modified waterborne coatings were presented as coating(x), where x denotes the pH values.

To test the electrochemical properties of the coatings, the polished 1 cm × 1 cm × 1 cm Q235 steel block was connected with a copper wire, which was encapsulated in a PVC ferrule with epoxy resin. Then, the steel block was coated with the prepared slurry by a high-pressure spray gun (w-71, ANEST IWATA Company, Shanghai, China) at room temperature, and dried in an oven at 50 °C for 1 h. The coating thickness was controlled at 50 ± 5 μm with a digital laser thickness gauge (QNIX4200, Rosengarten, Germany).

### 2.4. Characterization

(1)X-ray diffraction (XRD)

XRD patterns were collected on an X-ray diffractometer (D8 Advance, Bruker AXS GmbH, Karlsruhe, Germany) with Cu K_α_ (*λ* = 1.5418 Å) to identify the phase structures. All XRD spectra were measured in the range 2*θ* = 20–70° at a scanning speed of 2°/min. The average crystal size was calculated by the Scherrer equation.

(2)Transmission electron microscope (TEM) and energy dispersive spectroscopy (EDS)

The morphology, the element distribution, and the crystal structure were observed by a TEM (FEI Tecnai G2 F20, FEI Company, Portland, OR, USA) equipped with EDS at 200 kV.

(3)Fourier-transform infrared spectrometer (FT-IR)

The bond structure was measured by FTIR (Thermo Scientific Nicolet iS20, Waltham, MA, USA) in the range of 4000–400 cm^−1^ using a KBr pellet.

(4)Electrochemical properties

Electrochemical property detections were performed on an electrochemical workstation (Metrohm, Autolab, Utrecht, Switzerland) at room temperature. A platinum sheet (Pt) was used as the counter electrode, a saturated calomel electrode (SCE) was used as the reference electrode, and 3.5 wt.% NaCl aqueous solution was used as the electrolyte.

The polarization curve of α-Fe_2_O_3_ nanoparticles was detected as follows. The nanoparticles, liquid paraffin, and carbon powder were uniformly mixed with a ratio of 1:4:5 and worked as the working electrode. The measurements were performed at a scanning speed of 1 mV·s^−1^ within the range of −0.5 V–+2 V.

The polarization curve and electrochemical impedance (EIS) of the α-Fe_2_O_3_-modified waterborne coating are tested as follows. The coated steel block was selected as the working electrode, and the polarization curve was obtained in the range of −0.5 V–+1 V. The electrochemical impedance spectrum was scanned with 10mVRMS in the range of 0.01 Hz~100,000 Hz, which was further analyzed using NOVA 2.0 software (Metrohm, Beijing, China) to obtain the charge transfer resistance.

(5)Molecular dynamics simulation

Materials Studio (MS) software was used for the molecular dynamics’ simulation. The Visualizer module was used for crystal modeling. The universal force field was selected in the Forcite module to optimize the iron oxide lattice. The resulting energy-minimized lattice was dynamically simulated under the conditions of constant temperature (T), constant pressure (P), and the number of atoms (N) (NPT) ensemble with the temperature of 433 K, pressure of 1.20 MPa, and simulation time of 500 Ps.

## 3. Results and Discussion

### 3.1. La-Doped α-Fe_2_O_3_ Nanoparticle Characterization

#### 3.1.1. XRD

Figure 1 shows the XRD patterns of α-Fe_2_O_3_(u) and La-doped α-Fe_2_O_3_(x) (x = 4, 6, 8). Only α-Fe_2_O_3_ (JCPDS33-0664) is detected in all samples. The diffraction peaks attributed to LaO_x_ are not observed. For La-doped α-Fe_2_O_3_(4), La-doped α-Fe_2_O_3_(6), and La-doped α-Fe_2_O_3_(8), the diffraction peaks at 2q = 33.152° are shifted 0.024°, 0.044°, and 0.003° lower relative to JCPDS33-0664, respectively. In addition, the diffraction peaks of La-doped α-Fe_2_O_3_(4) are widened. These results illustrate that the large La^3+^ (radius = 0.106 nm) [32] is doped into the α-Fe_2_O_3_ lattice (Fe^3+^ radius = 0.064 nm) [33].

The average crystal size and the lattice constant of α-Fe_2_O_3_(u) and La-doped α-Fe_2_O_3_(x) (x = 4, 6, 8) are calculated by the Scherrer formula, as presented in Table 2. Obviously, with La doping at different pH values, the crystal size of α-Fe_2_O_3_ decreases and the lattice constant increases, of which La-doped α-Fe_2_O_3_(6) has the largest lattice constant and the largest lattice distortion.

#### 3.1.2. TEM, SAED, and TEM-EDS

Figure 2 shows the TEM images of α-Fe_2_O_3_(u) and La-doped α-Fe_2_O_3_(x) (x = 4, 6, 8). The α-Fe_2_O_3_(u) shows irregular nanospheres with weak agglomeration and an average particle size of 101.4 nm (Figure 2a). La-doped α-Fe_2_O_3_(4) shows irregular and weakly-agglomerated nanorods with a rough surface and an average particle length of 57.0 nm (Figure 2b). La-doped α-Fe_2_O_3_(6) shows irregular nanospheres with a smooth surface, which are 84.3 nm in diameter (Figure 2c). La-doped α-Fe_2_O_3_(8) shows agglomerated polygon morphology, which is not uniform, and the average size is 83.2 nm (Figure 2d). Clearly, with an increase in pH value, the average particle size first increases and then decreases, consistent with the change rule of the XRD results (Table 2). Compared to each other, La-doped α-Fe_2_O_3_(6) has the weakest agglomeration and better dispersion.

Figure 3 shows the high-resolution TEM images and selected area electron diffraction patterns of α-Fe_2_O_3_(u) and La-doped α-Fe_2_O_3_(x) (x = 4, 6, 8). For α-Fe_2_O_3_(u), the *d-spacing* value of 0.271 nm is assigned to the (104) crystal plane of α-Fe_2_O_3_ (Figure 3a). For La-doped α-Fe_2_O_3_(4), the *d-spacing* values of 0.374 nm and 0.273 nm are assigned to the (012) and (104) crystal plane of the α-Fe_2_O_3_, which are 0.006 nm and 0.003 nm larger than JCPDS33–0664, respectively (Figure 3c). For La-doped α-Fe_2_O_3_(6), the *d-spacing* values of 0.254 nm and 0.275 nm are assigned to the (012) and (104) crystal plane of the α-Fe_2_O_3_, which are 0.002 nm and 0.005 nm larger than JCPDS33–0664, respectively (Figure 3e). For La-doped α-Fe_2_O_3_(8), the *d-spacing* values of 0.372 nm and 0.272 nm are assigned to the (012) and (104) crystal plane of the α-Fe_2_O_3_, which are 0.004 nm and 0.002 nm larger than JCPDS33–0664, respectively (Figure 3g). These results further prove that the La element is doped into the α-Fe_2_O_3_ lattice, resulting in the α-Fe_2_O_3_ lattice expansion. In addition, compared with La-doped α-Fe_2_O_3_(x)(x = 4, 8), the increase in (104) crystal plane spacing is more obvious in La-doped α-Fe_2_O_3_(6), indicating La doped α-Fe_2_O_3_ (6) has the largest lattice distortion.

Moreover, as shown in Figure 3b,d,f,h, the α-Fe_2_O_3_(u) and La-doped α-Fe_2_O_3_(x) (x = 4,6,8) all have a polycrystalline structure, and the crystal planes of La_2_O_3_ are not observed, indicating that the La element is substituted in the α-Fe_2_O_3_ lattice.

Figure 4a–c shows the EDS mapping images of La-doped α-Fe_2_O_3_(x) (x = 4, 6, 8). Fe, O, and La elements appear in the same area, ascribed to the formation of a La-Fe-O solid solution where La is uniformly doped in the α-Fe_2_O_3_ lattice.

#### 3.1.3. Molecular Dynamics Simulation

Figure 5 shows the simulated crystal lattice of α-Fe_2_O_3_(u) and La-doped α-Fe_2_O_3_. Table 3 shows the simulated lattice constants, Fe-O bond length, and energy in α-Fe_2_O_3_(u) and La-doped α-Fe_2_O_3_. When the La element is doped in the α-Fe_2_O_3_ lattice, the lattice constant is increased by 0.170 Å, the average Fe-O bond length is decreased by 0.044 Å, and the Fe-O bond energy is increased by 185.699 kcal/mol. A La atom replaces the steric Fe sites and bonds to O, reducing the Fe-O bond length and enlarging the Fe-O bond energy, which improves the crystal stability [14,34].

#### 3.1.4. FT-IR

Figure 6 shows the FT-IR spectrum of α-Fe_2_O_3_(u) and La-doped α-Fe_2_O_3_(x) (x = 4, 6, 8). As presented in Figure 6a, the band at 3400 cm^−1^ is assigned to the stretching vibration of the hydroxyl group absorbed on the iron oxide surface [35]. Figure 6b shows the magnified view of the FTIR spectrum. As reported by Miah [36], the characteristic absorption peak of La_2_O_3_ appears at 574 cm^−1^. For La-doped α-Fe_2_O_3_(x) (x = 4, 6, 8), the bands at ~574 cm^−1^ have a redshift of 1.68 cm^−1^, 0.83 cm^−1^, and 0.90 cm^−1^, respectively. As reported by Wu [37], the characteristic absorption peak of Fe_2_O_3_ is around 464 cm^−1^. For α-Fe_2_O_3_(u), the curve shows only the Fe–O stretching vibration band at 470.01 cm^−1^. For La-doped α-Fe_2_O_3_(x) (x = 4, 6, 8), the bands at 470.01 cm^−1^ have a blueshift of 3.86 cm^−1^, 7.61 cm^−1^, and 3.58 cm^−1^, respectively. The lattice distortion caused by the La doping leads to a decrease in Fe–O bond length, i.e., an increase in the force constant. Therefore, the absorption peak of the Fe-O bond shifts to a higher wavenumber [38,39]. In addition, the blueshift in the FT-IR spectra of La-doped α-Fe_2_O_3_(6) is more obvious, indicating that the Fe-O bond is more distorted as the degree of lattice distortion increases.

### 3.2. Electrochemical Properties of La-Doped α-Fe_2_O_3_ Nanoparticles

Figure 7 shows the potentiodynamic polarization curves of α-Fe_2_O_3_(u) and La-doped α-Fe_2_O_3_(x) (x = 4, 6, 8). Corrosion potentials (*E*_corr_) and corrosion currents density (*I*_corr_) are applied to evaluate the corrosion resistance. The corrosion potentials of α-Fe_2_O_3_(u), La-doped α-Fe_2_O_3_(4), La-doped α-Fe_2_O_3_(6), and La-doped α-Fe_2_O_3_(8) are −41.10 mV, 2.45 mV, 28.52 mV, and −26.38 mV, respectively. The corrosion current densities of α-Fe_2_O_3_(u), La-doped α-Fe_2_O_3_(4), La-doped α-Fe_2_O_3_(6), and La-doped α-Fe_2_O_3_(8) are 0.013 μA/cm^−2^, 0.0024 μA/cm^−2^, 0.0013 μA/cm^−2^, and 0.0076 μA/cm^−2^, respectively. La-doped α-Fe_2_O_3_(6) has a relatively positive corrosion potential and a lower corrosion current density, indicating the higher corrosion resistance. The more difficult the bond destruction, the stronger the corrosion resistance [40]. Therefore, the higher Fe-O bond energy in La-doped α-Fe_2_O_3_(6) leads to its better corrosion resistance.

### 3.3. La-Doped α-Fe_2_O_3_ Modified Waterborne Coating

Figure 8 shows the potentiodynamic polarization curve of α-Fe_2_O_3_ and La-doped α-Fe_2_O_3_-modified waterborne coatings. The corrosion potentials of the coating(u), coating(4), coating(6), and coating(8) are −693.46 mV, −686.10 mV, −482.14 mV, and −706.65 mV, respectively. The corrosion current densities of the coating(u), coating(4), coating(6), and coating(8) are 0.28 μA/cm^−2^, 0.06 μA/cm^−2^, 0.028 μA/cm^−2^, and 1.26 μA/cm^−2^, respectively. Coating(6) has a relatively positive corrosion potential and a lower corrosion current density. Therefore, coating(6) has higher corrosion resistance.

Figure 9a shows the Nyquist plots of coating(u) and coating(x) (x = 4, 6, 8). Clearly, the capacitive arc radius of coating(6) is the largest. Based on the Nyquist plots, the corresponding impedance frequency diagram (Figure 9c) and phase frequency diagram (Figure 9d) are obtained. According to the literature [41], the impedance increases with an increase in the capacitive arc radius or the modulus; when the phase angle is close to 90°, the capacitive reactance is similar to the perfect capacitance, leading to a smaller capacitance and a larger impedance. Therefore, coating(6) has better corrosion resistance. Moreover, the equivalent circuit and *R_c_* values are obtained by fitting the Nyquist plots, as shown in Figure 10. *R_s_*, *R_c_*, Q, and W represent the electrolyte resistance, the coating charge transfer resistance, the coating double-layer capacitance, and the Warburg impedance, respectively. *R_c_* reflects the corrosion resistance. Coating(6) gives the largest *R_c_* value (1951.6 kΩ·cm^2^), followed by coating(4) (818.9 kΩ·cm^2^), coating(u) (69.8 kΩ·cm^2^), and coating(8) (28.4 kΩ·cm^2^), also indicating that the corrosion resistance of coating(6) is better.

Figure 11 shows the intersection Bode diagram (IBP) of coating(u) and coating(x) (x = 4, 6, 8). When the Bode phase plot and Bode impedance plot tend to be intersected at the upper left corner, the corrosion resistance is better. The corresponding phase angle and impedance of IBP show a similar increasing tendency, and the corresponding frequency shows a decreasing tendency [42]. Compared with these of coating(u) and coating(x) (x = 4, 8), the IBP value of coating(6) is the highest, indicating better corrosion resistance.

## 4. Conclusions

α-Fe_2_O_3_ and La-doped α-Fe_2_O_3_ were prepared using a hydrothermal process. The synthetic pH value affects the morphology and structure of La-doped α-Fe_2_O_3_. At pH = 4 and 8, the La-doped α-Fe_2_O_3_ nanoparticles show irregular morphology with agglomeration. At pH = 6, the La-doped α-Fe_2_O_3_ nanoparticles are well dispersed and show smooth nanospheres.

The α-Fe_2_O_3_ lattice is distorted by the La-doping. When the synthetic pH value is 6, the La doping leads to a larger lattice distortion, which strengthens the Fe–O bond in α-Fe_2_O_3_. The increase in binding energy improves the electrochemical stability of α-Fe_2_O_3_. As a result, the waterborne coating modified with La-doped α-Fe_2_O_3_ nanoparticles synthesized at pH = 6 has better corrosion resistance.

## Figures and Tables

**Figure 1 nanomaterials-12-03346-f001:**
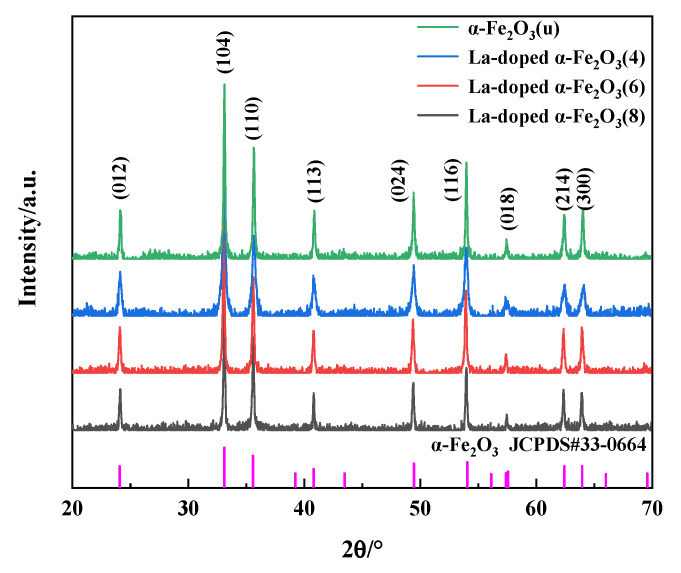
XRD patterns of α-Fe_2_O_3_(u) and La-doped α-Fe_2_O_3_(x) (x = 4, 6, 8).

**Figure 2 nanomaterials-12-03346-f002:**
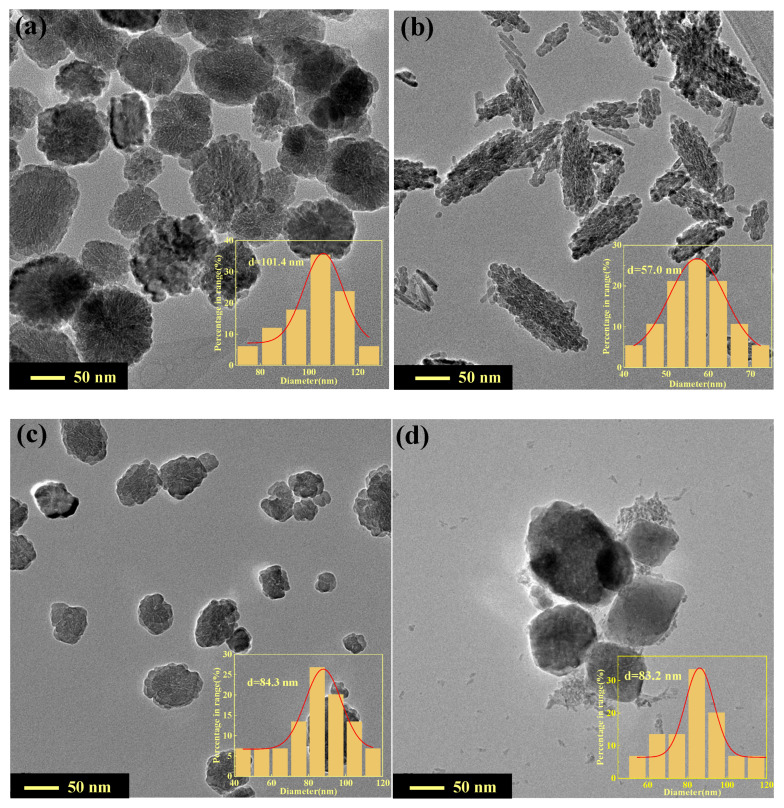
TEM images of α-Fe_2_O_3_(u) and La-doped α-Fe_2_O_3_(x) (x = 4, 6, 8). **a** α-Fe_2_O_3_(u); (**b**) La-doped α-Fe_2_O_3_(4); (**c**) La-doped α-Fe_2_O_3_(6); (**d**) La-doped α-Fe_2_O_3_(8).

**Figure 3 nanomaterials-12-03346-f003:**
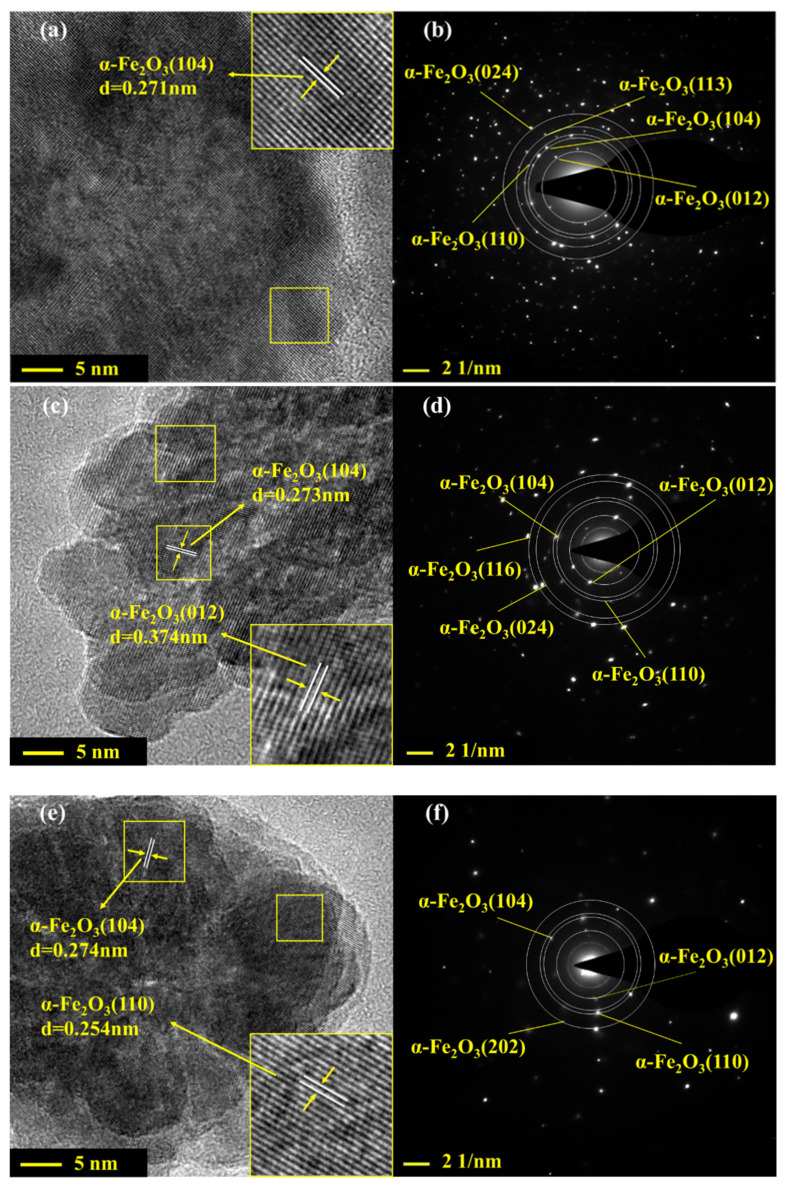
HRTEM and SAED images of α-Fe_2_O_3_(u) and La-doped α-Fe_2_O_3_(x) (x = 4, 6, 8). (**a**,**b**) α-Fe_2_O_3_(u); (**c**,**d**) La-doped α-Fe_2_O_3_(4); (**e**,**f**) La-doped α-Fe_2_O_3_(6); (**g**,**h**) La-doped α-Fe_2_O_3_(8).

**Figure 4 nanomaterials-12-03346-f004:**
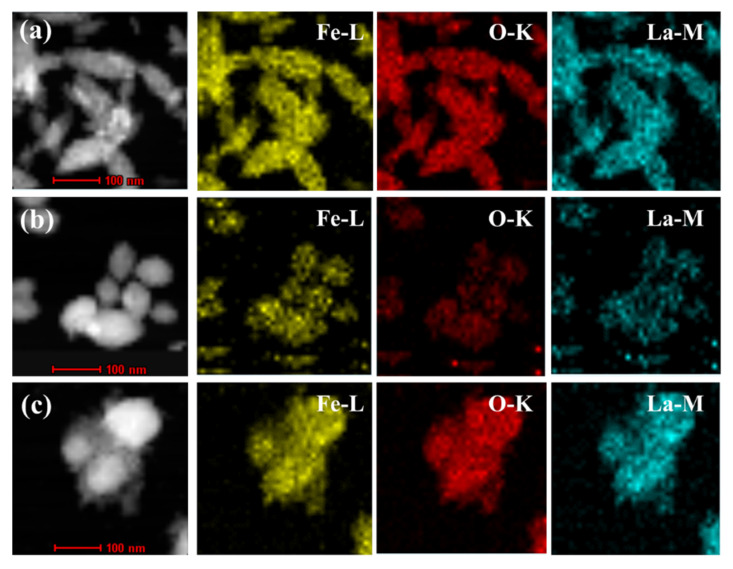
TEM-EDS mapping images of La-doped α-Fe_2_O_3_(x) (x = 4, 6, 8). (**a**) La-doped α-Fe_2_O_3_(4); (**b**) La-doped α-Fe_2_O_3_(6); (**c**) La-doped α-Fe_2_O_3_(8).

**Figure 5 nanomaterials-12-03346-f005:**
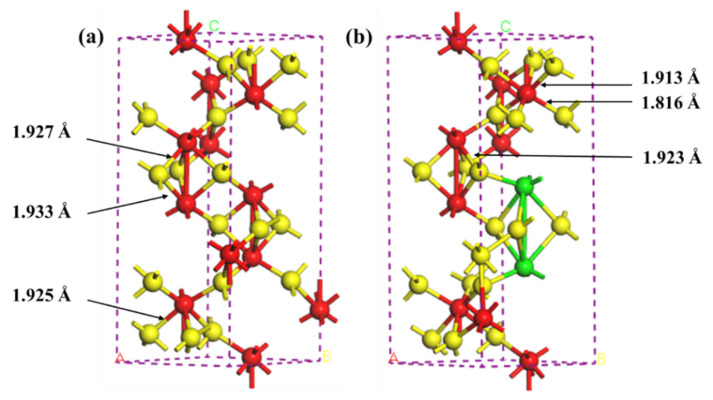
Crystal lattice of α-Fe_2_O_3_(u) or La-doped α-Fe_2_O_3_ (Yellow ball is O; red ball is Fe; green ball is La). (**a**) α-Fe_2_O_3_(u); (**b**) La-doped α-Fe_2_O_3_.

**Figure 6 nanomaterials-12-03346-f006:**
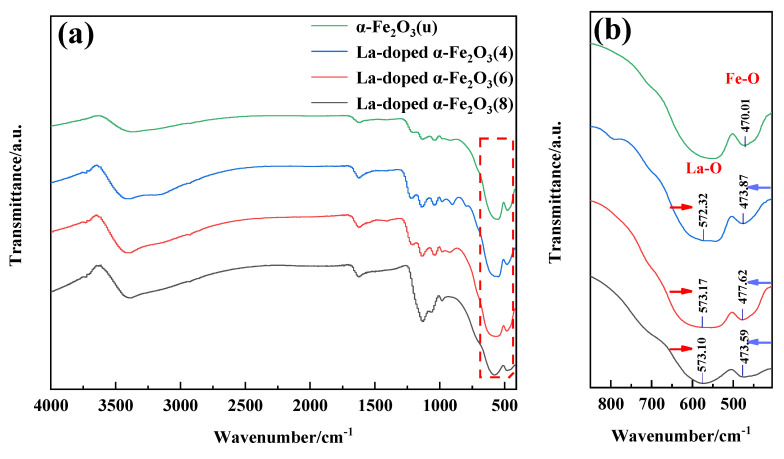
FT-IR spectra of α-Fe_2_O_3_(u) and La-doped α-Fe_2_O_3_(x) (x = 4, 6, 8), (**a**) 410~4000 cm^−1^; (**b**) 410~750 cm^−1^.

**Figure 7 nanomaterials-12-03346-f007:**
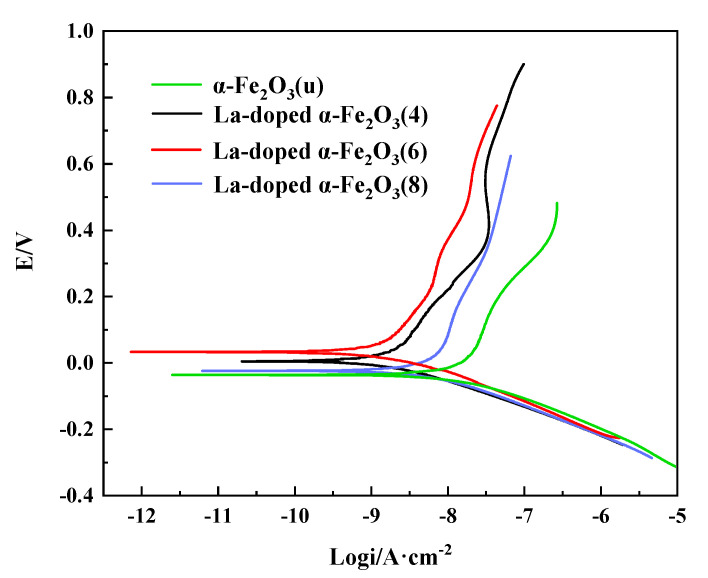
Polarization curves of α-Fe_2_O_3_(u) and La-doped α-Fe_2_O_3_(x) (x = 4, 6, 8).

**Figure 8 nanomaterials-12-03346-f008:**
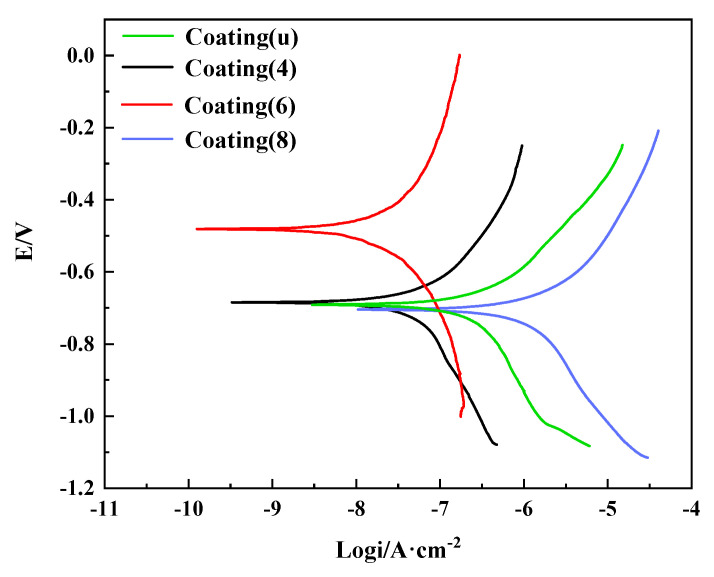
Potentiodynamic polarization curve of coating(u) and coating(x) (x = 4, 6, 8).

**Figure 9 nanomaterials-12-03346-f009:**
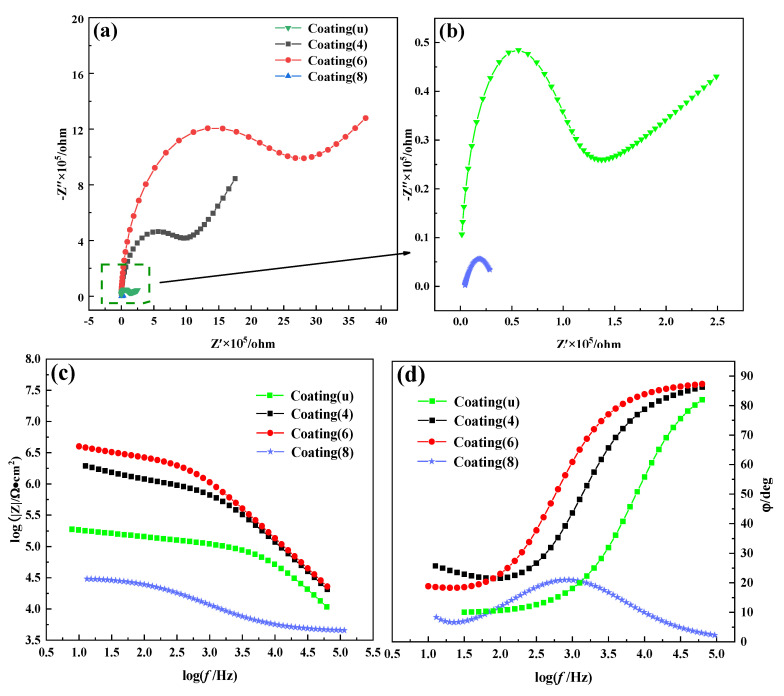
(**a**) Nyquist plots of coating(u) and coating(x) (x = 4, 6, 8); (**b**) magnified view of coating(u) and coating(8); (**c**) impedance frequency diagram of coating(u) and coating(x) (x = 4, 6, 8); (**d**) phase frequency diagram of coating(u) and coating(x) (x = 4, 6, 8).

**Figure 10 nanomaterials-12-03346-f010:**
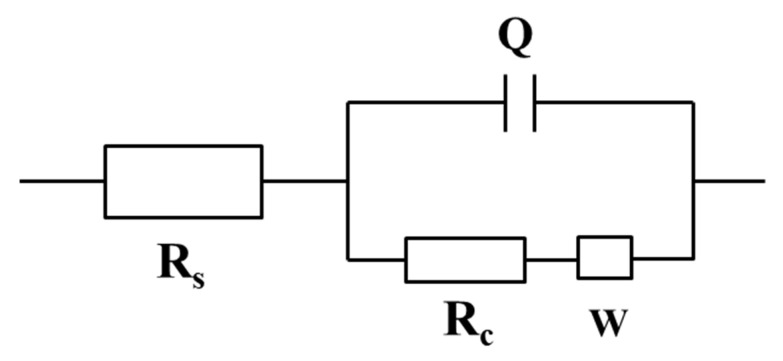
Equivalent circuit of the α-Fe_2_O_3_(u) coating and coating(x) (x = 4, 6, 8).

**Figure 11 nanomaterials-12-03346-f011:**
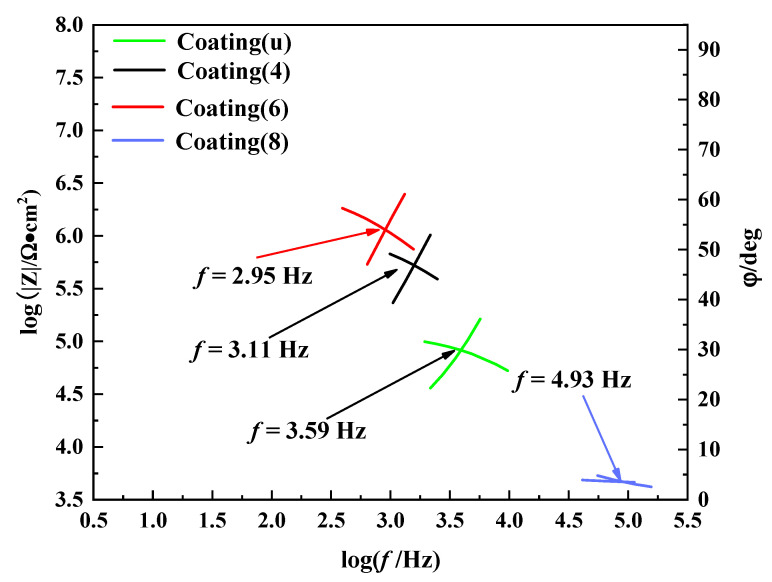
IBP of the coating(u) and coating(x) (x = 4, 6, 8).

**Table 1 nanomaterials-12-03346-t001:** Synthesis parameters of the La-doped α-Fe_2_O_3_-modified waterborne coatings.

Material	Feeding Amounts/g
H_2_O	58.0
Dispersing agent	0.3–0.4
Defoamer	0.2–0.3
Coalescent	3.0–3.2
Ethylene glycol	1.5–1.6
Iron oxide	18–30
Ammonia	0.3–0.4
Acrylic resin	66.0
Thickening agent	0.2–0.4

**Table 2 nanomaterials-12-03346-t002:** Average crystal size and lattice constant of nanoparticles.

Sample	Average Crystal Size/nm	Lattice Constant a/nm	Lattice Constant b/nm	Lattice Constant c/nm
α-Fe_2_O_3_(u)	99.5	0.50273	0.50273	1.37762
La-doped α-Fe_2_O_3_(4)	41.7	0.50351	0.50351	1.37696
La-doped α-Fe_2_O_3_(6)	83.3	0.50353	0.50353	1.37551
La-doped α-Fe_2_O_3_(8)	79.8	0.50344	0.50344	1.37672

**Table 3 nanomaterials-12-03346-t003:** Simulated lattice parameters.

	Lattice Parameters		
	Average Fe-O Bond Length/Å	Fe-O Bond Energy/(kcal·mol^−1^)	Lattice Constants a/Å
α-Fe_2_O_3_(u)	1.928	58.653	4.780
La-doped α-Fe_2_O_3_	1.884	244.352	4.950

## Data Availability

The study did not report any data.

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
