# Peer review of "Synthesis, Characterization, and Electrochemical Properties of La-Doped α-Fe2O3 Nanoparticles"

_nanomaterials, 2022, doi:10.3390/nano12193346_

Round 1

Reviewer 1 Report

The paper presents a synthesis of La-doped Fe2O3 nanoparticles. The presentation is clear and logical. Figure 9a is missing the axis label. The particles are referred to as microspheres and microrods in the text, when everything is nano. The English has room for improvement but the meaning is always clear.

Reviewer 2 Report

The authors reported the La-doped α-Fe2O3 nanoparticles for corrosion resistance. The characterizations support their conclusions solidly. So I recommended its publication in Nanomaterials after major revisions as follows:

1. What is the doping contents of La in Fe2O3. EDS or other characterizations may be needed to clarify that.

2. Why were the amounts of La2(SO4)3 and Fe2(SO4)3 fixed to be 1.7g and 120g? How was the ratio obtained? What will happen if more La2(SO4)3 are added for the reaction?

3. Three doped samples were obtained, and the sample prepared at pH of 6 has the best property. I was wondering what is the most important factor that determines the corrosion resistance, morphology, size, or doping content?

4. In line 156 page 4, the authors stated that “Clearly, with increase in pH value, the average particle size first increases and then decreases, consistent with XRD results”. That description seems not reasonable. Because the grain sizes are different from the crystal sizes.

5. The obtained materials are interesting with decent property. Is it possible to use them for the super-flexible devices for various applications, which is a frontier and hotspot at present. I suggest the authors analyze it with citing some closely related papers on Fe-based materials and flexible materials: 1) DOI: 10.1016/j.matt.2021.07.021; 2) DOI: 10.1007/s42765-022-00162-7; 3) DOI: 10.1002/advs.202103714.

6. There are many grammar errors in the manuscript, such as, line 23 in Page 1, environmentally-friendly should be environmental-friendly; line 27 in page 1, corrosion-resistance should be corrosion resistance. The authors should check the manuscript carefully and correct all of them.

Round 2

Reviewer 2 Report

After revision, the questions have been addressed, and the quality of the manuscript has been greatly improved. So I recommended its publication in Nanomaterials.